# Sustainable Carbon Derived from Sulfur-Free Lignins for Functional Electrical and Electrochemical Devices

**DOI:** 10.3390/nano12203630

**Published:** 2022-10-16

**Authors:** Bony Thomas, Mohini Sain, Kristiina Oksman

**Affiliations:** 1Division of Materials Science, Department of Engineering Sciences and Mathematics, Luleå University of Technology, SE-97187 Luleå, Sweden; 2Mechanical & Industrial Engineering (MIE), University of Toronto, Toronto, ON M5S 3G8, Canada; 3Wallenberg Wood Science Center (WWSC), Luleå University of Technology, SE-97187 Luleå, Sweden

**Keywords:** lignin, carbonization, microstructure, energy storage, electrical conductivity

## Abstract

Technical lignins, kraft, soda, lignoboost, and hydrolysis lignins were used for the production of carbon particles at different carbonization temperatures, 1000 °C and 1400 °C. The results showed that the lignin source and carbonization temperature significantly influenced the carbon quality and microstructure of the carbon particles. Soda lignin carbonized up to 1400 °C showed higher degree of graphitization and exhibited the highest electrical conductivity of 335 S·m^−1^, which makes it suitable for applications, such as electromagnetic interference shielding and conductive composite based structural energy storage devices. The obtained carbon particles also showed high surface area and hierarchical pore structure. Kraft lignin carbonized up to 1400 °C gives the highest BET surface area of 646 m^2^ g^−1^, which makes it a good candidate for electrode materials in energy storage applications. The energy storage application has been validated in a three-electrode set up device, and a specific capacitance of 97.2 F g^−1^ was obtained at a current density of 0.1 A g^−1^ while an energy density of 1.1 Wh kg^−1^ was observed at a power density of 50 W kg^−1^. These unique characteristics demonstrated the potential of kraft lignin-based carbon particles for electrochemical energy storage applications.

## 1. Introduction

Over the past decade, the development of biomass-based carbon materials and their conversion into different kind of carbon nanomaterials has gained immense attention in the research community and different biomass feedstocks have been converted into value added end products by researchers around the globe [1]. Out of different biomass feedstocks, one of the most promising feedstocks is lignin, which is an abundant biopolymer, waste product from paper and pulp industries with over 50 million metric tons isolated every year [2]. Lignin is pyrolyzed to get a carbon rich biochar which is used as soil amendment [2]. Recent studies have revealed that the lignin can be utilized to prepare high value-added carbon materials that can be used for different applications like conductive fillers for composites or electrodes for energy storage applications [1,2]. In addition to the compositional complexity of the feedstock, type of isolation technique plays a vital role in defining the final chemical structure of the technical lignins [3]. Kraft pulping, soda pulping, sulfite pulping, and organosolv pulping, etc. are the most common type of lignin isolation techniques [4]. Kraft and sulfite pulping are classified as sulfur bearing processes while organosolv and soda pulping are coming under sulphur free process. Kraft pulping uses a mixture of sodium hydroxide and sodium sulphide known as white liquor while soda pulping uses only sodium hydroxide [4]. One of the recent advancements in the kraft pulping process is the lignoboost process, which provides more efficient extraction with high purity lignin, higher yield, low amounts of ash and lower carbohydrate content [5]. Hydrolysis lignin is another type of lignin obtained as a byproduct from the bio-fuel industries when the lignin containing biomass is treated with hydrolytic enzymes [6]. Different carbon materials, including carbon nanofibers [7,8], carbon aerogels [5,9], and carbon particulates [10], have been produced from extracted lignins using different isolation processes. To make carbon nanofibers, researchers are following different processing techniques like melt spinning, solution spinning and electrospinning [7]. Recently, carbon aerogels were prepared using ice templating followed by freeze drying and carbonization [5,9]. Direct carbonization has been performed to make lignin-based carbon particles as a replacement for carbon black [10]. Different activation strategies [11] have been performed to increase the properties of lignin derived carbon materials which are not environment friendly and cost effective [12]. Metal catalyzed hydrothermal carbonization has been performed for the preparation of graphitic biocarbon from lignin [13]. Even though most these studies have shown that a specific type of lignin can be used for preparing carbon materials with specific properties, there is a need for a comprehensive study where the properties of carbons from most common technical lignins can be compared at different carbonization temperatures (without any chemical or physical activation and in the absence of any catalysts). 

In the current study, four different commonly available technical lignins, namely kraft lignin (KL), soda lignin (SL), lignoboost lignin (LB), and hydrolysis lignin (HL), were carbonized at two different carbonization temperatures 1000 °C and 1400 °C. A simple, green, and low-cost strategy such as a direct carbonization of the lignin, was adopted. Neither activation steps nor any special templating agents, techniques, additional additives, or extra processing steps were used to enhance the porosity or microstructure of the carbon particles. Resulting carbon particles were systematically analyzed for their suitability for high value applications, such as supercapacitor electrodes and conductive graphitic carbon additives. The carbon particles with the highest specific surface area were used for making supercapacitor electrodes and their electrochemical performances were analyzed. Carbon particles were also evaluated for their electrical conductivities to determine technical lignin and temperature of carbonization for achieving highest degree of graphitization for their effective use as a conductive filler. 

Remarkable electrochemical and electrical properties were achieved, and the resulting graphitic carbon obtained in this study is another important step forward towards achieving functional graphitic carbon from biomass. Hence, this study presents the importance of types of lignin and their carbonization process, and how they impact on the final properties of carbon materials. This work gives new insights towards the effective utilization and value addition of an abundant and unique natural resource, reducing the carbon footprint, economic growth potential, and improving ecosystem sustainability.

## 2. Experimental

### 2.1. Materials

Four different lignins were used; (1) a low sulfonate content kraft lignin (KL) supplied from Sigma-Aldrich (St. Louis, MI, USA); (2) soda lignin (SL) Protobind 2000, from Greenvalue Enterprises LLC, (Media, PA, USA); (3) Lignoboost lignin, Biochoice^TM^ lignin (LB), based on softwood was kindly supplied by Domtar Plymouth pulp mill (Plymouth, NC, USA) and (4) hydrolysis lignin (HL), which was kindly supplied by a Finnish energy company, St1 biofuels (Kajaani, Finland). Sulphuric acid (H_2_SO_4_, 95–98%) and ethanol was purchased from VWR International AB (Stockholm, Sweden). Polytetrafluoroethylene (PTFE) (60 wt.% aqueous suspension) was purchased from Sigma-Aldrich (Stockholm, Sweden).

### 2.2. Preparation of Carbon Particles

Before the carbonization, all the lignin powders were powdered and sieved using an 80 μm sieve size to the similar particle size (<80 μm). A horizontal tube furnace Nabertherm RHTC-230/15, Nabertherm GmbH (Lilienthal, Germany) was used for the carbonization in nitrogen (N_2_) atmosphere. The setup used was as follows: first one hour at 100 °C, then one hour at 400 °C, and then two hours at the final temperatures 1000 °C or 1400 °C. All the heating steps during the carbonization process were performed at a rate of 5 °C/min. After carbonization, the samples were cooled down to room temperature and taken out carefully for further characterizations. Table 1 shows the different carbons and their samples codes. 

### 2.3. Electrode Preparation

Working electrode for three electrode measurement was prepared by mixing prepared carbon particles, polytetrafluoroethylene (PTFE), and carbon black in ethanol, in the weight ratio 75:15:10, where PTFE acted as polymeric binder and carbon black as a conductive filler. This slurry was then left for 6 h in oven at 100 °C to evaporate ethanol (which was used to disperse the carbon particles and PTFE in the slurry) and water (from the aqueous suspension of PTFE). After drying, the working electrode was weighed (Approximately 20 mg) and used for the electrochemical measurements. 

### 2.4. Characterization

TGA-Q500 (TA Instruments, New Castle, DE, USA) was used for the thermo gravimetric analysis (TGA) in nitrogen (N_2_) atmosphere, from room temperature to 1000 °C, maintaining a heating rate of 10 °C/min. 

Equation (1) was used for the calculation of carbon yield,
(1)$$\mathrm{Carbon}\;\mathrm{yield}=\frac{\mathrm{mass}\;\mathrm{of}\;\mathrm{carbon}\;\mathrm{obtained}\;}{\mathrm{mass}\;\mathrm{of}\;\mathrm{lignin}\;\mathrm{initally}\;\mathrm{taken}\;}\times 100$$
where the mass of the lignin before and mass of carbon after carbonization was measured for the determination of the yield.

The microstructure of the carbon particles was studied using scanning electron microscopy (SEM) JSM-IT300, (JEOL, Tokyo, Japan) with an acceleration voltage of 20 kV under a high vacuum. The samples preparation was performed by diluting aqueous dispersions of the carbon particles to approximately 0.01 wt.%, the dispersion was drop casted onto a freshly cleaved mica sheet attached to the sample holder, and the dispersion was dried before sputter coating with platinum to a thickness of 15 nm. 

The elemental analysis of the carbon materials was carried out using energy dispersive X-ray spectroscopy (EDX) on the same SEM instrument equipped with a silicon drift detector Oxford X-MaxN 50 mm^2^, (Oxford Instruments, Abingdon, UK). 

The particle sizes were determined from SEM images taken at random locations of the samples. For each sample, the sizes of at least 200 particles were measured by the software ImageJ (University of Wisconsin, Madison, WI, USA). 

A Bruker Senterra Raman microscope, (Bruker Corporation, Billerica, MA, USA), using green laser (*λ* = 532 nm) with a power of 5 mW, was used to study carbon structure of the prepared materials. The measurements were performed at a magnification of 20×, and the spectra were recorded from 50 to 3500 cm^−1^ with an acquisition time of 2 s. 

X-Ray diffraction (XRD) was performed with a PANalytical EMPYREAN diffractometer (Malvern Instruments, Malvern, UK) using Cu K alpha radiation. Intensity data were recorded in intervals of 0.03° between the scattering angles 2*θ* = 5° and 50°. Small angle X-ray scattering was carried out with a PixCel3D detector and a graphite monochromator. Based on the full width at half maximum (FWHM), the XRD crystallite size was calculated. Thus, the radial expansion of the carbon crystal and the stacking height of graphene layers were determined using the Scherrer equation, as shown in Equation (2) [14].
(2)$$L_{i}=\frac{K_{i}\lambda \;}{\beta_{i}\cos\theta_{i}\;},\;i=a,\;c\;$$
where *L_i_* (nm) can take two indices: *L_a_* which is the radial expansion of the carbon crystal contributed by the (001) peak and *L_c_* which is the stacking height of the graphene layers can be determined from (002) peak. *K_i_* is a structural constant and values used were *K_a_* = 1.84 and *K_c_* = 0.89 taken from the literature [15]. *λ* is the wavelength of *Cu K_α_* radiation, *λ* = 0.1542 nm [15]. Finally, *β_i_* (rad) is the FWHM for each peak and *θ_i_* (rad) is the diffraction angle for each peak. 

The distance (*d*) between two graphene layers was calculated using Bragg’s law for *n* = 1,
(3)$$2d\sin\theta_{c}=n\lambda$$

The number of graphene layers per stack unit was calculated by dividing the stacking height (*L_c_*) with interplanar distance (*d*). 

Specific surface area (SSA), average pore size, and pore volume of the carbon particles were measured using Micromeritics Gemini VII 2390a Brunauer–Emmett–Teller (BET) analyzer at 77 K (Micromeritics Instrument Corporation, Norcross, GA, USA). The samples were degassed at 300 °C for 3 h prior the BET measurement in a Micromeritics FlowPrep060 to remove entrapped moisture and increase the accessibility to the pores. Micropore area was calculated using the standard t-plot method and total pore volume was determined from the amount of N_2_ adsorbed at a relative pressure P/P_0_ = 0.99. 

Electrochemical properties of KL1400 electrode were measured using a Princeton Applied Research VerstaSTAT 3 Potentiostat/Galvanostat (AMETEK Scientific Instruments, Wokingham, UK) connected with a three-electrode cell kit (Pine Research Instrumentation, Durham, NC, USA). The working electrode was prepared as described in the previous section. Platinum was used as the counter electrode, Ag/AgCl as reference electrode and the electrolyte used was 1M sulphuric acid (H_2_SO_4_) solution. To analyze the rate capabilities of the electrodes, cyclic voltammetry measurements (CV) were carried out at different scan rates between 2 and 100 mV s^−1^ in the potential range from 0 to 1 V. Specific capacitances obtained from CV tests can be calculated using Equation (4),
(4)$$C=\frac{1}{mv\left(V_{2}-V_{1}\right)}\int_{V_{1}}^{V_{2}}IdV$$
where *C* (F g^−1^) is the specific capacitance; *m* (mg) is the mass of active materials loaded in the working electrode; *v* (V s^−1^) is the scan rate; *I* (A) is the discharge current; *V*_2_ and *V*_1_ (V) are high and low potential limits of the CV tests. Further galvanostatic charge discharge method (GCD) was conducted using at different current densities in the range from 0.1 to 1 Ag^−1^ to investigate the electrochemical performance. Specific capacitance from GCD measurements can be calculated using Equation (5),
(5)$$C=\frac{I∆t}{m∆V}$$
where *C* (F g^−1^) is the specific capacitance; *I* (A) is the discharge current; ∆*t* (s) is the discharge time; ∆*V* (V) is the potential window; and *m* (mg) is the total mass of electrode material. 

To calculate the specific energy density (*E*) and specific power density (*P*), a practical supercapacitor was assembled using the prepared electrodes and two electrode method was used for the measurements and the details of which is given in supporting information. From the galvanostatic tests, *E* and *P* can be calculated using Equations (6) and (7),
(6)$$E=\frac{C∆V^{2}}{2*3.6*4}$$
(7)$$P=\frac{E*3600}{∆t}$$
where *E* (Wh kg^−1^) is the average energy density; *C* (F g^−1^) is the specific capacitance; ∆*V* (∆*V*) is the potential window; *P* (W kg^−1^) is the average power density and ∆*t* (s) is the discharge time. Electrochemical impedance spectroscopy (EIS) measurements were carried out in the frequency range of 10^−2^ to 10^5^ Hz to determine the resistances offered by the electrode. 

The electrical conductivity of the carbon particles was measured at room temperature using a Hioki IM 3536 LCR meter (Hioki E.E. Corporation, Nagano, Japan). The conductivity values were recorded for frequencies ranging from 1 kHz to 5 MHz.

## 3. Results and Discussions

Figure 1a shows the schematic of the preparation process of carbon particles from different technical lignin. Kraft (KL), soda (SL), lignoboost (LB), and hydrolysis lignin (HL) powders were carbonized in a horizontal tubular furnace maintained in N_2_ atmosphere to 1000 and 1400 °C to obtain the different carbon particles as listed in Table 1. 

Figure 1b shows the results obtained from thermogravimetric analysis (TGA) for different lignins. KL resulted in highest amount of char residue, 45% at 1000 °C. LB and HL lignins showed approximately same amount of char residue, 38% while SL had lowest percentage of char residue of 31%. To analyze in more depth the thermal decomposition behaviors, the derivative thermogravimetric (DTG) curve for all lignins has been plotted and shown in Figure 1c. All the lignins had their major decomposition peaks (DTG_max_) in the temperature range between to 300 °C to 400 °C, as shown in Table 2, which corresponds to the degradation of lignin. SL and LB showed slight decomposition below 200 °C which corresponds to the early release of volatiles and SL released much more volatiles than LB, which could be the reason for lower amount of residue for SL in comparison with all other lignins. KL and HL had no decomposition peaks and hence no release of volatiles for temperatures less than 200 °C. Smaller decomposition peaks observed between 200 °C and 250 °C for SL, LB, and HL (SL—250 °C, LB—219 °C, HL—221 °C) could be due to the presence hemicellulose. Above 600 °C, lignins produce aromatics and CO_2_ as major decomposition products during TGA and KL showed relatively stronger peaks between 600 °C and 700 °C, indicating the release of more CO and CO_2_ [9] which could result in carbon particles with high specific surface area (SSA) compared to other lignins. In the TG-IR studies conducted by Zhang et al., it has been observed that KL produced relatively smaller amounts of volatiles, and no carbonyl groups were released during the temperature range below 1000 °C, leading to approximately 46% char residue [16]. 

Experimental carbon yield obtained after carbonization of lignin particles are represented in Figure 1d. It was observed that the carbon yields were higher at 1000 °C compared to that at 1400 °C and the values for 1000 °C were in accordance with the amount of residue obtained from TGA analysis. At 1400 °C, KL and SL had almost similar carbon yield (KL—31% and SL—30%). KL showed a significant reduction in carbon yield (from 43% at 1000 °C to 31% at 1400 °C) with increase in temperature, indicating the release of more carbon as volatiles, which could lead to the formation of numerous micropores (<2 nm) which results in higher SSA and hierarchical pore structure. LB and HL did not show remarkable differences for carbon yields between 1000 °C and 1400 °C. It has been reported by Liu et. al. [17] that the first step of the depolymerization of lignin chain is the breaking of the β-O-4 linkage. Collision between these free radicals potentially leads to the chemical bond formation leading to formation of stable compounds, and the random bonding of the radicals results in polyaromatic biochar for temperatures higher than 350 °C [17]. This indicates all lignins formed stable carbons already during the isothermal heating period at 1000 °C. The heat energy supplied during the isothermal processing at 1400 °C for 2 h could be attributed to the reorganization of the carbon structure to develop graphitic domains. The later proposed mechanism yet to be confirmed. The carbon content and molecular weight (MW) of the lignins can affect the final carbon yield. In one of our previous studies, Wei et al. [18] reported that HL and SL had considerably higher overall ratios of aliphatic to aromatic carbons than KL and LB. KL and HL have similar MW and showed similar carbon yield [18], which is in conformity to the results discussed here.

The morphology of carbon particles was analyzed using scanning electron microscopy (SEM) and the results are shown in Figure 2. KL1000 and KL1400 showed rough surface texture compared to all other lignins which showed flat and smooth surfaces. Comparing KL1000 and KL1400 the surface morphology did not change noticeably while SL1400 clearly showed a smoother and layered structure compared to SL1000. LB1000 had smooth surface while well-defined layered structure was visible for LB1400. HL showed similar surface for HL1000 and HL1400, but it was observed that many smaller carbon particles were attached to the bigger sized particles. A direct comparison of particle size was difficult in the case of SL, LB, and HL because the lignin powder melted and fused together to form bigger agglomerates which were broken down to uniform size using a mortar and pestle. KL retained particle morphology even after carbonization, which is confirmed using the particle size measurements, performed qualitatively using ImageJ. KL particles (average diameter was 5.33 µm) experienced a volume shrinkage during the carbonization process (average diameter for KL1000 was 4.3 µm, and for KL1400 2.2 µm). Thus, it can be concluded that KL1000 and KL1400 show relatively rougher surfaces with nanostructured wrinkles and surface cavities. The latter can provide higher SSA compared to all other lignins at the same time as SL and LB and is expected to have higher degrees of graphitization compared to other lignins, which is yet to be confirmed.

Elemental analysis of carbon particles was performed using energy dispersive X-ray spectroscopy (EDX) and the results obtained are listed in Table 3. The atomic percentage of carbon was between 86% and 97% for all carbon particles and the percentage of carbon was increased with increase in the carbonization temperature. The amount of oxygen was varied between 3% to 12% between the carbon particles. The atomic percentage of oxygen was decreased with the increase in carbonization temperature indicating the breakage of more oxygen containing bonds utilizing the higher supplied energy, and more carbonaceous gases, such as CO and CO_2_, were released. Minor amounts of sodium (Na) and sulfur (s) were present in KL and SL based carbon particles, which were derived from the lignin isolation process. Minor amounts of potassium (K) and silicon (Si) were present in SL based carbon particles. These were due to the impurities from the wheat straw arising from the starting materials for isolating soda lignin. The presence of ash also can contribute to the presence of silicon in the carbonized lignins. A High purity of lignoboost lignin [5] could be the reason for THE presence of very small amounts of inorganic elements in LB1000 and LB1400. HL lignin carbon particles were also free from inorganic elements which might have been released as volatiles during the carbonization. It was observed that at higher carbonization temperatures, there was no considerable difference in elemental percentages detected by different elemental analysis techniques, such as SEM-EDX and XPS. [19]. EDX provides bulk elemental compositions because of the higher X-ray generation depth of around 1 µm, while the results from the XPS are more at the surface, within the depth around 10 nm. Since the bulk elemental composition is more important for electrochemical properties, SEM-EDX analysis is expected to provide reliable information about the bulk composition of carbon particles [19]. 

Raman spectra of carbon particles are represented in Figure 3. All carbon particles showed characteristic bands for carbon materials at 1582 cm^−1^ (G-band) and at 1330 cm^−1^ (D-band). Vibrations of sp^2^ bonded carbon atoms in the 2D hexagonal lattice is the reason for G-band and in-plane terminated disordered tangling bonds in graphite, as represented by the D-band [15,20]. Raman spectra are also evaluated using the intensity ratio between D and G bands (I_D_/I_G_ ratio) [20,21,22]. 

Figure 4 shows the I_D_/I_G_ ratio and full width at half maximum (FWHM) of D-band of the carbon particles carbonized at 1000 °C and 1400 °C calculated from the Raman spectra. In Figure 4a, it is seen that I_D_/I_G_ ratio increased significantly when the temperature was increased from 1000 °C to 1400 °C. This clearly indicates the growth of aromatic clusters and graphitic layers in the carbon structure [15,21,23]. In addition to the increase, the shape of Raman spectra changed significantly with the increase in temperature. To analyze the changes in full width at half maximum (FWHM) and band positions with respect to carbonization temperature, simple curve fitting involving only D and G bands [24] was used (Figure 4b). During curve fitting, D bands were fitted using the Gaussian function and G bands were fitted with the Lorentzian function, which is in accordance with literature [24]. For all carbon particles the D-band became narrower as can be seen in Figure 4b, the values are given in Table 4. It is also seen that the valley intensity between D and G bands was also decreased (Figure 3). The FWHM for G bands (Table 4) was increased when the carbonization temperature was increased except for HL. This could be due to the conversion of aromatic rings to small graphite crystallite when the carbonization temperature was increased [24]. These results indicate the increase of ordering in the carbon structure during the high temperature carbonization as the non-crystalline carbon was evolved as volatiles [24]. 

The positions of D and G bands are given in Table 5. The D and G bands showed a red shift with the increase in carbonization temperature. The shift in D band peak position was prominent in case of KL, SL and LB based carbon particles while the position didn’t change for HL based carbon particles. The red shift in the peak position of D band also demonstrates the increase in graphitization due to the formation of larger aromatic clusters which agree with the biomass-based chars reported in the literature [24,25,26]. The red shift observed in the G band peak position could be due to the introduction of more defective aromatic rings into the already generated graphite layers as the carbonization temperature was increased [22]. A similar observation of red shift of G band peak has been observed in the nanocrystalline carbon and the presence of defects and impurities in the microstructure led to anharmonic contribution to the lattice potential, which resulted in the red shift [27]. All the carbon particles except HL1400 showed more defined 2D peaks than the broad 2D peak observed at 1000 °C. It is important to note that the parameters, such as peak positions of D and G bands and FWHM of D and G bands, can differ significantly with respect to the biomass feedstock [24]. The chemical structure of the biomass such as combination of aromatic rings and the connections between different chemical groups vary significantly for each lignin which can affect the final structure of the carbon [24]. 

XRD patterns of carbon particles were recorded and represented in Figure 5. For all carbon particles produced at 1000 °C the first diffraction peak was observed between 2*θ* = 22–23° and the second peak was located between 2*θ* = 43–44°. The first diffraction peak (2*θ* = 22–23°) corresponds to the (002) plane which represents the reflections from stacked graphene layers. The second maximum (2*θ* = 43–44°) corresponds to the (100) plane, which originates from the aromatic ring structures present within the graphene layers [15]. With the increase in the carbonization temperature, the peak intensities of both the planes were increased and the peaks became more prominent at 1400 °C. This indicated the growth of graphitic layers with the carbonization temperature. Among all carbon particles at 1400 °C, SL1400 was found to have more prominent (002) and (100) peaks, showing the higher degree of graphitization, as indicated by the lowest FWHM of the Raman D band as shown in Figure 4b. 

To quantitatively analyze the (002) and (100) peaks, Bragg’s Law (*n* = 1) and Scherrer-Equation were used and mean distance between graphene layers d, stacking height *L_c_* and radial expansion *L_a_* were calculated as shown in Table 6. It has been observed that the radial expansion *L_a_* increased with increase in the carbonization temperature and the highest expansion (*L_a_* = 5.65 nm and *L_c_* = 1.25 nm) was observed for SL1400, which indicated the higher level of graphitization achieved, as discussed earlier. The distance between the stacked layers were calculated and the value is ranging between 0.377 nm–0.395 nm (Table 6). This is in accordance with the results obtained in the graphitization analysis undertaken by Schneider et al. [15], where the biochar derived from the high temperature pyrolysis of beech wood showed similar inter layer distance of 0.387 nm at 1600 °C. The lowest interlayer distance was found for SL1400 (0.377 nm), which is close to the pure graphite which has 0.354 nm as the interlayer distance between the layers and which can be achieved for temperatures above 2100 °C [15,28]. The number of layers in the stack was calculated by dividing *L_c_* value by the interlayer distance d [15]. It was observed that the stacks consist of approximately 2–3 layers in all the carbon particles. A similar observation of constant stack height and stack order has been reported in the literature and the carbonization temperature needs to be increased to make a noticeable difference in these parameters [15,28,29]. Overall, XRD analysis confirmed the graphitization occurred during the high temperature carbonization of lignin particles. 

The specific surface area (SSA), average pore diameter (d_a_), and pore volume (V_p_) of the carbon particles were measured using Brunauer–Emmett–Teller (BET) analysis. Nitrogen (N_2_) adsorption isotherms for all the samples carbonized at 1000 °C and 1400 °C are represented in Figure 6a,b. Detailed information about the microstructure of carbon particles is shown in Table 7. Except for HL, all the carbon particles produced at 1400 °C exhibited higher SSA than those produced at 1000 °C, as shown in Figure 6c. This can be attributed to the generation of more micropores since more and more carbon was eliminated as volatiles during the high temperature carbonization. KL1000 showed highest SSA of 266 m^2^ g^−1^ among the samples produced at 1000 °C while KL1400 showed 646 m^2^ g^−1^ among the samples carbonized at 1400 °C. 

The pore volume of carbon particles is graphically represented in Figure 6d. Pore volume followed a similar tendency to SSA with KL1400 having the highest pore volume of 0.31 cm^−3^.This indicates that KL is generating more volatiles during both carbonization temperatures, leading to the generation of a more porous microstructure with average pore size of 1.89 nm, which could be suitable for electrochemical applications.

LB (71 m^2^ g^−1^ at 1000 °C and 151 m^2^ g^−1^ at 1400 °C) and SL (81 m^2^ g^−1^ at 1000 °C and 216 m^2^ g^−1^ at 1400 °C) showed quite similar adsorption behavior during the BET analysis, but SL showed the lowest adsorption and hence lowest SSA among all the samples for both carbonization temperatures. This could be due to the higher graphitization and the growth of crystallites [30] in comparison with other carbon particles, as discussed in the Raman and XRD analysis. HL1400 showed lower SSA than HL1000 and it was observed that the SSA contributed by micropores was reduced in HL1400 (157 m^2^ g^−1^) compared to HL1000 (290 m^2^ g^−1^) while the contribution from meso and macropores remained almost the same for both HL1000 and HL1400. To gain a better understanding of the SSA, pore size distributions of all carbon particles are shown in Appendix A. In addition to the higher contribution of SSA from micropores (<2 nm), KL1400, which has highest surface area among all carbon particles, showed a higher contribution of SSA from pore sizes ranging from 2 to 6 nm (Appendix A) compared to others.

The electrochemical properties of the KL1400 based electrode were studied using cyclic voltammetry (CV) and galvanostatic charge discharge (GCD) measurements using a three-electrode system with 1M sulphuric acid (H_2_SO_4_) solution as the electrolyte. CV measurements were carried out at different scan rates and the obtained results are shown in Figure 7a. Near rectangular shaped CVs were observed at lower scan rates and slight deviation from rectangular shapes were observed at high scan rates, indicating the good rate capability of the KL1400 electrode. 

At 2 mV s^−1^, the specific capacitance observed was 151 F g ^−1^ while the specific capacitance was 16 F g^−1^ at the scan rate of 100 mV s^−1^ which is proportional to the area under the cyclic voltammograms shown in Figure 7a. Galvanostatic charge discharge measurements were carried out for different current densities and the results are shown in Figure 7b. The highest specific capacitance obtained was 97.2 F g ^−1^ at a current density of 0.1 A g^−1^. Even at 1 A g^−1^, electrode retained a specific capacitance of 45 F g ^−1^, which showed the capability of electrode material to store energy even at higher current densities. Figure 7c shows the specific capacitance values obtained from the GCD measurements. 

Appendix A shows the GCD curves for the practical supercapacitor measured using 2 electrode method and Appendix A shows the obtained specific capacitance values. At 0.05 A g^−1^, the supercapacitor made up of KL1400 electrodes showed a specific capacitance of 50 F g^−1^. The highest energy density obtained was 1.7 Wh kg^−1^ at a power density of 25 W kg^−1^, while at a higher power density of 50 W kg^−1^ the energy density was 1.1 Wh kg^−1^. Electrochemical impedance spectroscopy (EIS) has been carried out in the frequency range between 10^−2^ and 10^5^ Hz and the Nyquist plot for KL1400 is shown in Figure 7d. KL1400 exhibited negligible equivalent series resistance (<1 Ω) and low charge transfer resistance (5.4 Ω). The lower charge transfer resistance indicates the lesser resistance experienced by the electrolytes in the pores of the electrodes and better contact between electrode and current collector [31]. The nearly vertical portion in the Nyquist plot (Figure 7d) at the low frequency region indicates the proper infiltration of pores with the electrolyte ions. Thus, the KL1400 electrode exhibited ideal capacitative behavior at the low frequencies. The Ragone plot for KL1400 is shown in Figure 8a. All these results indicate the suitability of KL1400 to be used as electrodes in supercapacitors. Electrical conductivity is the primary material property of concern when using the carbon particles as conductive reinforcements in composites for energy storage or electromagnetic interference shielding applications [32,33,34]. 

Figure 8b shows the electrical conductivity values obtained for all the carbon particles at a frequency of 1 kHz. It has been observed that with the increase in the carbonization temperature the electrical conductivity values were increased. This is in accordance with the increase in the graphitization observed during the high temperature carbonization, which was also proved using Raman spectroscopy and XRD analysis. SL1400, which showed highest degree of graphitization, showed the highest electrical conductivity of 335 S m^−1^ while KL1000 showed the lowest electrical conductivity of the order of 10^−6^ S m^−1^. The direct comparison of electrical conductivity values between different studies is difficult due to different factors, e.g., the difference in lignin resource, carbonization procedures, and conductivity measurement conditions, such as the equipment and conditions of the measurement. However, the conductivity values obtained for SL1400 is much higher than the conductivity values reported by Snowden et al. [10] for the ball milled carbonized lignin (0.9 S m^−1^) and that of lignin-based carbon fillers (142 S m^−1^) reported by Gindl-Altmutter et al. [35]. Table 8 illustrates the superior electrical conductivity of SL1400 compared to other carbon materials produced by direct carbonization of lignin. In conclusion, SL based carbon particles showed a higher degree of electrical conductivities at both carbonization temperatures, indicating the suitability of SL for making conductive carbon particles for the preparation of conductive additives for composites. 

## 4. Conclusions

In the current study, carbon particles were prepared from four different types of technical lignins. Two different final carbonization temperatures were chosen to study the impact of carbonization temperature on the microstructural evolution. It has been concluded that, except for hydrolysis lignin, the specific surface area was increased with the increase in the carbonization temperature. This is attributed to the evolution of more volatiles at higher temperatures. Kraft lignin carbonized at 1400 °C showed the highest surface area and was used as electrode material together with PTFE as binder to show promising electrochemical properties and hence suitability as electrodes in supercapacitor. It was also observed that the extent of graphitization was influenced by the type of lignin and carbonization temperature. Soda lignin at 1400 °C showed the highest electrical conductivity and could be utilized as functional carbon additives in composites, e.g., in EMI shielding applications. Hence, the current study presents new insights regarding the importance of the proper selection of technical lignins and their carbonization conditions for achieving the requirements of specific applications. 

## Figures and Tables

**Figure 1 nanomaterials-12-03630-f001:**
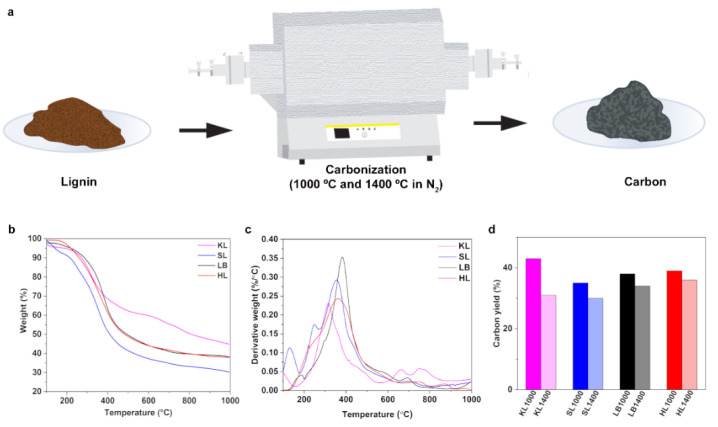
(**a**) Schematic showing the preparation of lignin-based carbon particles using the horizontal carbonization furnace. (**b**) Thermal behavior of and residues obtained for lignins at 1000 °C after the TGA analysis in N_2_ atmosphere. (**c**) Derivative thermogravimetric (DTG) curve for lignins. (**d**) Carbon yield obtained after carbonization at 1000 °C and 1400 °C for different lignins used in the study.

**Figure 2 nanomaterials-12-03630-f002:**
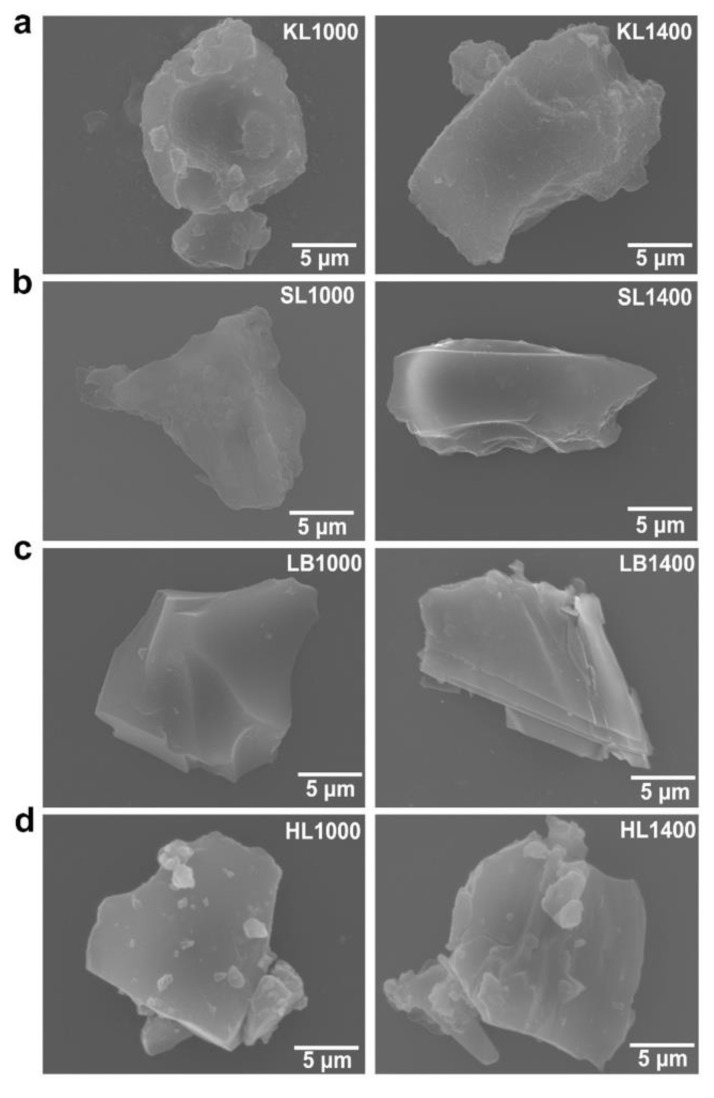
Microstructure of lignin-based carbon particles at 1000 °C and 1400 °C respectively (**a**) KL1000 and KL1400, (**b**) SL1000 and SL1400, (**c**) LB1000 and LB1400, (**d**) HL1000 and HL1400.

**Figure 3 nanomaterials-12-03630-f003:**
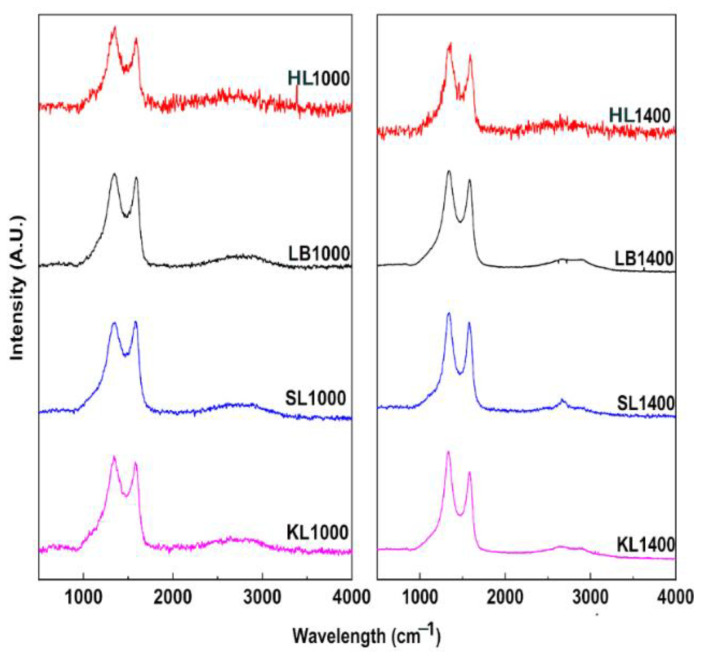
Raman spectra of carbonized HL, LB, SL and KL at 1000 °C and 1400 °C respectively.

**Figure 4 nanomaterials-12-03630-f004:**
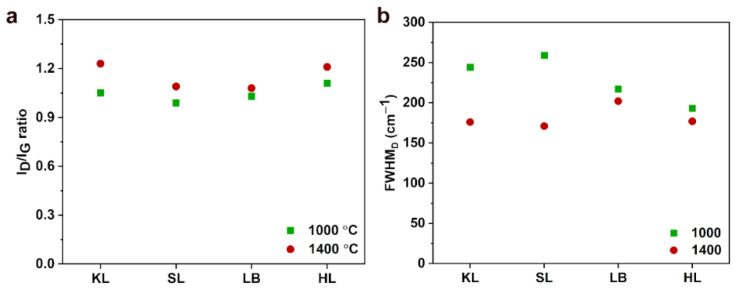
(**a**) I_D_/I_G_ ratio of carbon particles carbonized at 1000 °C and 1400 °C and calculated from the Raman spectra. (**b**) Full width at half maximum (FWHM) of D-band for KL, SL, LB and HL carbon particles calculated from the Raman spectra shown in Figure 3.

**Figure 5 nanomaterials-12-03630-f005:**
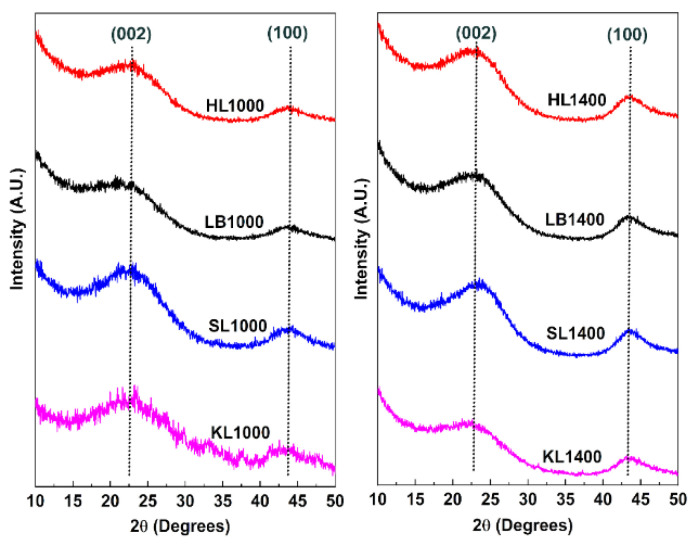
X- ray diffraction patters for KL, SL, LB, HL carbonized at 1000 °C and 1400 °C.

**Figure 6 nanomaterials-12-03630-f006:**
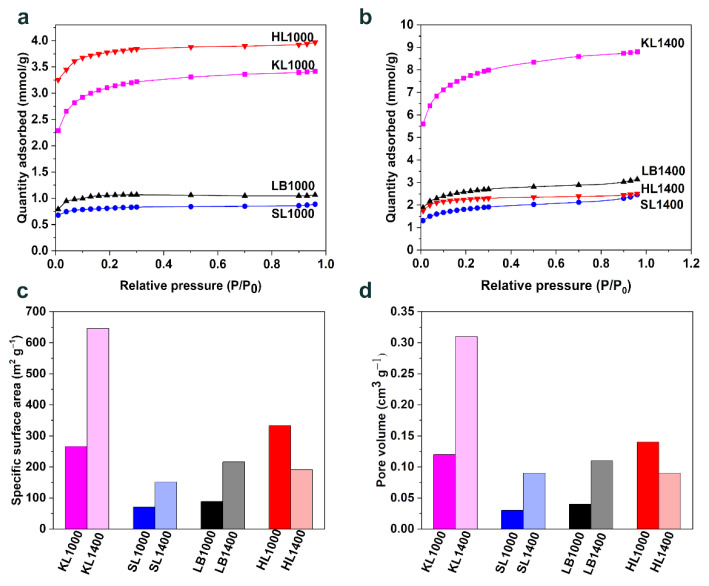
Results obtained from BET analysis (**a**,**b**) Nitrogen adsorption isotherms for carbon particles produced at 1000 °C and 1400 °C respectively, (**c**) BET specific surface area for all carbon particles, (**d**) Comparison of pore volumes obtained for all carbons.

**Figure 7 nanomaterials-12-03630-f007:**
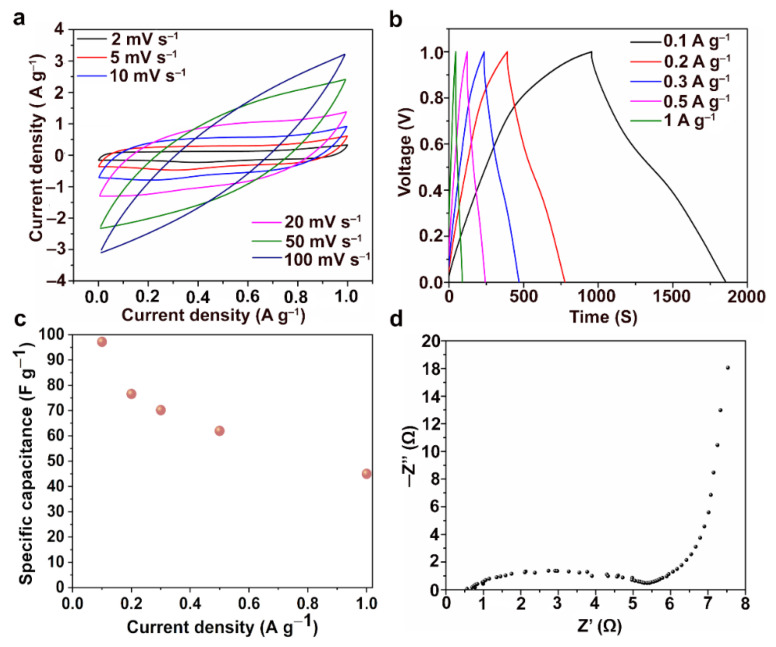
Results from the electrochemical analysis performed for KL1400 electrode. (**a**) Cyclic voltammograms (CVs) at different scan rates; (**b**) Galvano-static charge discharge (GCD) diagrams at different current densities; (**c**) specific capacitance values obtained at different current densities and (**d**) Nyquist plot obtained from electrochemical impedance analysis.

**Figure 8 nanomaterials-12-03630-f008:**
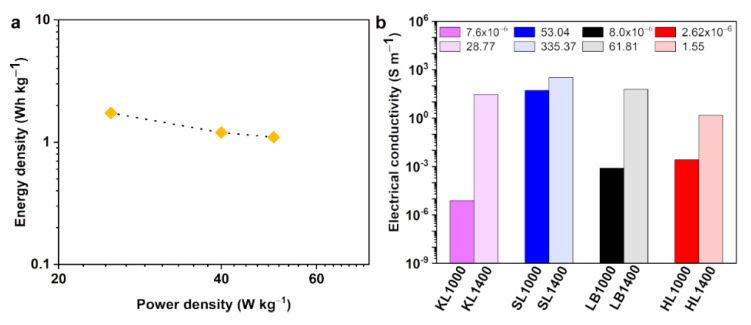
(**a**) Ragone plot for KL1400 electrode showing the energy density at different power densities and (**b**) Electrical conductivity values obtained for all carbon particles.

**Table 1 nanomaterials-12-03630-t001:** Table showing the type of lignin, carbonization temperature and the sample code for the prepared carbons.

	Carbonization Temperature (°C)	Sample Codes
Kraft lignin (KL)	1000	KL1000
	1400	KL1400
Soda lignin (SL)	1000	SL1000
	1400	SL1400
Lignoboost lignin (LB)	1000	LB1000
	1400	LB1400
Hydrolysis lignin (HL)	1000	HL1000
	1400	HL1400

**Table 2 nanomaterials-12-03630-t002:** Table showing the thermal properties of used materials, DTG_max_ and percentual amount of the char residue, obtained from TGA.

	DTG Max (°C)	Char Residue (%)
KL	315	45
SL	355	38
LB	383	38
HL	360	31

**Table 3 nanomaterials-12-03630-t003:** Elemental composition in at.% of different lignins carbonized at 1000 and 1400 °C obtained from SEM-EDX analysis.

Sample	C	O	Na	S	Si	K
KL1000	86.0	12.1	0.30	0.90	0.70	-
KL1400	88.0	9.90	0.40	0.20	1.50	-
SL1000	96.3	3.10	0.10	0.10	0.30	0.10
SL1400	94.7	4.40	0.40	0.40	0.10	-
LB1000	92.3	7.00	0.10	0.30	0.20	0.10
LB1400	96.5	3.50	-	-	-	-
HL1000	90.4	9.60	-	-	-	-
HL1400	96.9	3.10	-	-	-	-

**Table 4 nanomaterials-12-03630-t004:** Full width at half maximum (FWHM) of D and G Raman bands for carbon particles with respect to carbonization temperatures.

Sample	D_FWHM_ (cm^−1^)	G_FWHM_ (cm^−1^)	Sample	D_FWHM_ (cm^−1^)	G_FWHM_ (cm^−1^)
KL1000	244.3 ± 1.6	84.9 ± 0.9	KL1400	176.1 ± 1.1	92.4 ± 0.8
SL1000	259.7 ± 2.0	85.7 ± 1.0	SL1400	171.4 ± 1.1	87.7 ± 0.7
LB1000	217.8 ± 1.0	85.9 ± 0.6	LB1400	202.3 ± 1.2	90.1 ± 0.7
HL1000	193.6 ± 1.6	90.3 ± 1.1	HL1400	177.4 ± 1.3	89.1 ± 1.0

**Table 5 nanomaterials-12-03630-t005:** Temperature dependence of peak positions for D and G bands obtained from Raman spectroscopy analysis.

Sample	D Band Position (cm^−1^)	G Band Position (cm^−1^)	Sample	D Band Position (cm^−1^)	G Band Position (cm^−1^)
KL1000	1350	1582	KL1400	1331	1575
SL1000	1351	1582	SL1400	1337	1576
LB1000	1347	1584	LB1400	1339	1579
HL1000	1347	1584	HL1400	1347	1583

**Table 6 nanomaterials-12-03630-t006:** Results of quantitative analysis of XRD spectrum. Values for radial expansion (*L_a_*), stacking height (*L_c_*) and distance between graphene layers (d) obtained for all carbon particles at 1000 °C and 1400 °C.

Sample	*L_a_* (nm)	*L_c_* (nm)	d (nm)	Sample	*L_a_* (nm)	*L_c_* (nm)	d (nm)
KL1000	4.10	1.05	0.387	KL1400	5.46	1.13	0.387
SL1000	4.75	1.06	0.387	SL1400	5.65	1.25	0.377
LB1000	3.04	0.99	0.395	LB1400	4.33	1.12	0.382
HL1000	3.32	1.05	0.385	HL1400	4.18	1.08	0.384

**Table 7 nanomaterials-12-03630-t007:** SSA provided by micropores, contribution towards the SSA by meso and macropores, average pore diameter and pore volume obtained from BET surface area analysis.

Sample	SSA (m^2^ g^−1^)	S_micro_ (m^2^ g^−1^)	S_meso+macro_ (m^2^ g^−1^)	d_a_ (nm)	V_p_ (cm^3^ g^−1^)
KL1000	266	186	80	1.79	0.12
KL1400	646	414	232	1.89	0.31
SL1000	71	58	13	1.74	0.03
SL1400	151	88	63	2.25	0.09
LB1000	89	70	19	1.65	0.04
LB1400	216	136	80	2.01	0.11
HL1000	333	290	43	1.65	0.14
HL1400	191	157	34	1.78	0.09

(SSA—specific surface area, S_micro_—micropore area, d_a_—average pore diameter, V_p_—pore volume).

**Table 8 nanomaterials-12-03630-t008:** Comparison of electrical conductivity values of lignin-based carbon particles produced by direct carbonization.

Sample	Carbonization Temperature	Electrical Conductivity	Reference Number
SL1400	1400 °C	335 S m^−1^	This study
Carbonized lignin particles	900 °C	0.9 S m^−1^	[10]
Carbon fillers	2000 °C	142 S m^−1^	[35]
L-900	900 °C	33.3 S m^−1^	[36]

## Data Availability

The data presented in this study will be available on reasonable request from the corresponding author.

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
