# Peer review of "Sustainable Carbon Derived from Sulfur-Free Lignins for Functional Electrical and Electrochemical Devices"

_nanomaterials, 2022, doi:10.3390/nano12203630_

Round 1

Reviewer 1 Report

This work systematically investigated the effect of the lignin resource and carbonization temperature on the carbon quality and microstructure of the carbon particles, which provided new insights into the selection of technical lignins for achieving the requirements of specific applications. Therefore, I am pleased to recommend publication after the authors have resolved the following issues:

 1. In the introduction, the authors may wish to consider citing recent findings on the lignin-derived carbons. (Chemical Engineering Journal 392 (2020) 123721; Industrial Crops & Products 174 (2021) 114184; Journal of Colloid and Interface Science 628 (2022) 90–99; Carbon 196 (2022) 819827)

 2. The elemental analysis of the four technical lignins should be included since the carbon content is an important factor affecting carbon yield.

 3. In the experimental section, whether the carbons have been etched by acid? The metal impurities may affect the application performance of the carbons.

 4. In Figure 3, the degree of graphitization of the carbon decreases with increasing temperature. The author needs to explain the reasons for this anomaly in detail.

 5. In Figure 6, the pore size distribution curves should be included. KL1400 shows a significantly improved SSA, what is the reason?

 6. For the electrochemical analysis, it makes no sense to calculate the energy density and power density based on the three-electrode cell. The authors show carry out the test based on the two-electrode configuration. Moreover, the Equation 7 should be “P=3600E/t”.

Author Response

Comments to the Author 

This work systematically investigated the effect of the lignin resource and carbonization temperature on the carbon quality and microstructure of the carbon particles, which provided new insights into the selection of technical lignins for achieving the requirements of specific applications. Therefore, I am pleased to recommend publication after the authors have resolved the following issues:

Respected reviewer, we are thankful for your positive and constructive feedback. We are grateful to have this opportunity to improve the quality of our manuscript accordingly, and we have revised the paper addressing the comments. The revision is shown with a red color.

Comment:  In the introduction, the authors may wish to consider citing recent findings on the lignin-derived carbons. (Chemical Engineering Journal 392 (2020) 123721; Industrial Crops & Products 174 (2021) 114184; Journal of Colloid and Interface Science 628 (2022) 90–99; Carbon 196 (2022) 819–827)

Response: Thank you for your valuable suggestions. We have read these interesting articles, however, all deals with chemical modification of lignin to make a composite of lignin such as LS/ZnC2O4, lignin/silica, lignin/F127/ZnC2O4 etc. The aim of this study was to analyze the carbonized lignin alone, without any modifications or additives. We greatly appreciate the suggestions and believe that these articles are giving new insights for our future research, but we feel that these do not suit as references in this study.

Comment: The elemental analysis of the four technical lignins should be included since the carbon content is an important factor affecting carbon yield.

Response: Thank you for the comment. Authors agree that carbon content affects the carbon yield in the lignin. In one of our previous studies, we performed NMR for analyzing the chemical structure of all four lignin samples [Wei et.al, ACS Sustain Chem Eng 2021, 9, 12142–12154]. This semiquantitative analysis suggested that HL and SL had considerably higher overall ratios of aliphatic to aromatic carbons than KL and LB. Changes have been made in the manuscript on page number 6, line 236 to 240 as given below

“Carbon content and molecular weight (MW) of the lignins can also affect the final carbon yield and in one of our previous studies, Wei et. al [18] reported that HL and SL had considerably higher overall ratios of aliphatic to aromatic carbons than KL and LB.  KL and HL which have similar MW showed similar carbon yield [18] which is in accordance with the results discussed here.”

Comment:  In the experimental section, whether the carbons have been etched by acid? The metal impurities may affect the application performance of the carbons.

Response: Thank you, we have not washed or etched the carbon particles with any acids or chemicals. The quantity of metallic elements was negligible. Also, there was no signs of redox reactions in the cyclic voltammetry and therefore we do believe that there was no influence from the metallic impurities during the electrochemical analysis.

Comment:  In Figure 3, the degree of graphitization of the carbon decreases with increasing temperature. The author needs to explain the reasons for this anomaly in detail.

Response: Thank you for the observation. We agree that the relative intensity of G band is reduced compared to D band when the carbonization temperature was increased. Which is also observed in different carbon materials from lignin [2]. But that does not indicate the graphitization of the carbon is decreasing with increase in temperature. Increase in the graphitization is proved in the manuscript using the Id/Ig ratio, narrowing of and separation of D and G bands, FWHM or D and G bands, relative positioning of these peaks. All these facts are supported using relevant literatures. Increase in graphitization is further evidenced using XRD measurements and calculation of crystallite size. We believe that adequate information and explanations for proving the increase in graphitization with increase in carbonization temperature has already been included in the manuscript. Page number 8, line 296 to 313,

“It was observed that the ID/IG ratio was increased significantly when the temperature was increased from 1000 °C to 1400 °C (Figure 4). This clearly indicates the growth of aromatic clusters and graphitic layers in the carbon structure.  In addition to the increase in the ID/IG ratio as shown Figure 4(a), the shape of Raman spectra changed significantly with increase in temperature. To analyze the changes in full width at half maximum (FWHM) and band positions with respect to carbonization temperature, simple curve fitting involving only D and G bands was used. During curve fitting, D bands were fitted using gaussian function and G bands were fitted with Lorentzian function which is in accordance with literature. For all carbon particles the D-band became narrower as can be observed in Figure 4(b), whose values are given in Table 3. It can also be observed that the valley intensity between D and G bands was also de-creased (Figure 3). The FWHM for G bands (Table 3) were increased when the carbonization temperature was increased except for HL. This could be due to the conversion of aromatic rings to small graphite crystallite when the carbonization temperature was increased [24]. These results indicate the increase of ordering in the carbon structure during the high temperature carbonization as the non-crystalline carbon was evolved as volatiles.”

Page number 10, line 328-330.

“The red shift in the peak position of D band is also demonstrating the increase in graphitization due to the formation of larger aromatic clusters which agree with the bi-omass based chars reported in the literature”

Comment:  In Figure 6, the pore size distribution curves should be included. KL1400 shows a significantly improved SSA, what is the reason?

Response: Thank you for this very valuable comment. We have added the pore size distribution for the carbon particles in the supporting information (Figure S1). From the pore size distribution KL1400 found to have much higher contribution of micropores compared to other carbon particles which is the reason for higher BET SSA for KL1400. Following change has been made in the main text, page number 12, line 410 to 414 “To have better understanding of the SSA, pore size distributions of all carbon parti-cles are shown in Figure S1 a&b. KL1400 which has highest surface area among all carbon particles showed higher contribution of SSA from pore size ranging from 2 to 6 nm compared to others.”

Comment:  For the electrochemical analysis, it makes no sense to calculate the energy density and power density based on the three-electrode cell. The authors show carry out the test based on the two-electrode configuration. Moreover, the Equation 7 should be “P=3600E/t”.

Response: Thank you so much for correcting the mistake. We have performed two-electrode measurement and the corresponding values of energy density and power density have been added in the manuscript. Equation 7 is also changed accordingly. Following changes were made, Page number 5, line 175 to 180, “To calculate the specific energy density (E) and specific power density (P) a practical supercapacitor was assembled using the prepared electrodes and two electrode method was used for the measurements and the details of which is given in supporting information.  From the galvanostatic tests E and P can be calculated from Equation 6 and Equation 7,

E= (C∆V^2)/(2*3.6*4)          (6)

P=(E*3600)/∆t          (7) “

Page number 13, line 439 – 444

“Figure S2a shows the GCD curves for the practical supercapacitor measured using 2 electrode method and Figure S2b shows the obtained specific capacitance values. At 0.05 A g─1 supercapacitor made up of KL1400 electrodes showed a specific capacitance of 50 F g─1. Highest energy density obtained was 1.7 Wh kg─1 at a power density of 25 W kg─1 while at a higher power density of 50 W kg─1 the energy density was 1.1 Wh kg─1.”

Details of two electrode measurement is added in the supporting information along with Figure S2a and b.

Reviewer 2 Report

In this paper, "Sustainable carbon for electrochemical energy storage and electrical conductivity", the authors investigated the physicochemical properties of different technical lignin, which showed excellent electrochemical performances for electrochemical energy storage (specific capacitance of 97.2 F g‒1 at 0.1 A g‒1). Finally, a high value utilization of different lignin species was achieved. However, this paper still has some errors and needs major revisions, and my suggestions are as follows:

1.      Using SEM-EDX analysis to test the elemental content may be less precise and other methods are recommended for quantitative analysis. The reason is that the elemental content may have a greater influence on the electrochemical performance, such as, O element.

2.      Can the equations for calculating the electrical conductivity be written in the manuscript?

3.      Is it accurate to use Raman and XRD to calculate electrical conductivity results? It is recommended to use other methods of calculation.

4.      As stated in the article " The later application has been validated in a three-electrode set up device and a specific capacitance of 97.2 F g1 was obtained at a current density of 0.1 A g1 while an energy density of 13.5 Wh kg1 was observed at a power density of 50 W kg1." You have tested the power density and energy density using three electrodes, which is inaccurate. This is because the three electrodes can only be used to test the properties of a single working electrode, and cannot be used as a complete device to calculate energy density and power density. You need to make a symmetrical device to test the energy density and power density will be more accurate.

5.      The title is not accurate that can not reflect the research topic of supercapacitor

6.      The origin of the specific surface area should be discussed.

Author Response

Responses to Reviewer #2:

In this paper, "Sustainable carbon for electrochemical energy storage and electrical conductivity", the authors investigated the physicochemical properties of different technical lignin, which showed excellent electrochemical performances for electrochemical energy storage (specific capacitance of 97.2 F g‒1 at 0.1 A g‒1). Finally, a high value utilization of different lignin species was achieved. However, this paper still has some errors and needs major revisions, and my suggestions are as follows:

Respected reviewer, we have thankful for the time and efforts you have used to improve the quality of our paper. The paper is revised addressing the comments and changes are shown with a red color.

Comment:  Using SEM-EDX analysis to test the elemental content may be less precise and other methods are recommended for quantitative analysis. The reason is that the elemental content may have a greater influence on the electrochemical performance, such as, O element.

Response: Thank you for the valuable comment, we agree that elemental composition play a vital role in the electrochemical performance. In one of the previous studies, we compared the difference between EDX and XPS analysis and found that there is not much difference in the elemental compositions of different elements. It was also proved that at high temperatures like 1400 °C the difference become even negligible. XPS and EDX has different analysis capabilities with respect to the penetration depth. XPS is more surface analysis while EDX can be used for obtaining bulk elemental composition [Wei et.al, Front Mater 2019, 6]. Since the electrochemical performance is not only depends on the surface characteristics of the carbon particles authors believe that bulk elemental analysis using EDX would be appropriate.  

Following change has been made in the main text on page number 8, lines 281 to 288.  “It has been observed that at higher carbonization temperatures there is no considerable difference in elemental percentages detected using different elemental analysis techniques such as SEM-EDX and XPS analysis. EDX provide bulk elemental compositions because of the higher X-ray generation depths of around 1 µm while for XPS the results are more in surface level with depth up to 10 nm. Since the bulk elemental composition is more important for electrochemical properties, SEM-EDX analysis gives reliable information about the bulk composition of carbon particles.”

Comment:  Can the equations for calculating the electrical conductivity be written in the manuscript?

Response: Thank you for the comment. As mentioned in the manuscript, electrical conductivity of the carbon particles was measured at room using a Hioki IM 3536 LCR meter (Hioki E.E. Corporation, Nagano, Japan). There is no equation given in the manual of the instrument. Instrument directly displays the conductivity values.

Comment: Is it accurate to use Raman and XRD to calculate electrical conductivity results? It is recommended to use other methods of calculation.

Response: Thank you for the comment, we want to point out that no Raman data or XRD results were used for the calculation of electrical conductivity. As mentioned in the previously conductivity measurement was done using an LCR meter which has an inbuilt function to do the calculations and provide the conductivity values.

Comment: As stated in the article " The later application has been validated in a three-electrode set up device and a specific capacitance of 97.2 F g−1 was obtained at a current density of 0.1 A g−1 while an energy density of 13.5 Wh kg−1 was observed at a power density of 50 W kg−1." You have tested the power density and energy density using three electrodes, which is inaccurate. This is because the three electrodes can only be used to test the properties of a single working electrode and cannot be used as a complete device to calculate energy density and power density. You need to make a symmetrical device to test the energy density and power density will be more accurate.

Response: Thank you for the valid suggestion. We have performed the two-electrode measurement and the energy density and power density results are modified based on that.

Following changes were made, Page number 5, line 175 to 180,

“To calculate the specific energy density (E) and specific power density (P) a practical supercapacitor was assembled using the prepared electrodes and two electrode method was used for the measurements and the details of which is given in supporting information.  From the galvanostatic tests E and P can be calculated from Equation 6 and Equation 7,

E= (C∆V^2)/(2*3.6*4)          (6)

P=(E*3600)/∆t          (7) “

Page number 13, line 439 – 444

“Figure S2a shows the GCD curves for the practical supercapacitor measured using 2 electrode method and Figure S2b shows the obtained specific capacitance values. At 0.05 A g─1 supercapacitor made up of KL1400 electrodes showed a specific capacitance of 50 F g─1. Highest energy density obtained was 1.7 Wh kg─1 at a power density of 25 W kg─1 while at a higher power density of 50 W kg─1 the energy density was 1.1 Wh kg─1.”

Details of two electrode measurement is added in the supporting information along with Figure S2a and b.

Comment: The title is not accurate that cannot reflect the research topic of supercapacitor

Response: Thank you. Title has been modified to ‘Sustainable carbon for electrochemical supercapacitors and electrical conductivity’

Comment: The origin of the specific surface area should be discussed

Response: Thank you for the suggestion. Pore size distributions for all the carbon particles have been added in the supporting information (Figure S1) which would provide more insights into the specific surface area of carbon particles. Corresponding discussion has been added into the manuscript.

Following change has been made in the main text, page number 12, line 410 to 414 “To have better understanding of the SSA, pore size distributions of all carbon particles are shown in Figure S1 a&b. In addition to the higher contribution of SSA from micropores (< 2 nm), KL1400 which has highest surface area among all carbon particles showed higher contribution of SSA from pore size ranging from 2 to 6 nm (Figure S1 a&b) compared to others.

Reviewer 3 Report

I have carefully read this paper entitled with “Sustainable carbon for electrochemical energy storage and electrical conductivity". its interesting work, as a result, I have only a few minor points that the authors should address before it is accepted for publication. Please, publish subject to the following revisions:

1-      Rewrite the novelty statement at the end of the introduction section.

2-      Authors should justify the importance of the current work of how it is different from earlier reports. So, it’s better to add comparison table material and its performance to show the importance of the manuscript.

3-      Please add the EIS analysis (Nyquist plot) for different samples.

Author Response

Responses to Reviewer #3:

I have carefully read this paper entitled with “Sustainable carbon for electrochemical energy storage and electrical conductivity". its interesting work, as a result, I have only a few minor points that the authors should address before it is accepted for publication. Please, publish subject to the following revisions:

Respected reviewer, we are grateful for your valuable suggestions and improving the quality of our paper. The paper is modified addressing the comments and changes are marked with a red color.

Comment:  Rewrite the novelty statement at the end of the introduction section.

Response: Thank you for the suggestion, we have modified some parts of the introduction section to improve the clarity about the novelty. Following change was made, page number 2, line number 64 to 83,

“In the current study, four different commonly available technical lignins, kraft lignin (KL), soda lignin (SL), lignoboost lignin (LB), and hydrolysis lignin (HL) were carbonized at two different carbonization temperatures 1000 ºC and 1400 ºC. Extremely simple, green, and low-cost preparation strategy, direct carbonization of the lignins, was adopted. No activation steps, no special templating agents or techniques, no additional additives and no extra processing steps were used to enhance the porosity or microstructure of the carbon particles. Resulting carbon particles were systematically analyzed for their suitability for high value applications, such as supercapacitor electrodes and conductive graphitic carbon additives. The carbon particles with the highest specific surface area were used for making supercapacitor electrodes and their electrochemical performances were analyzed. Carbon particles were also evaluated for their electrical conductivities to determine which technical lignin and temperature of carbonization would provide highest degree of graphitization and hence can be used as conductive filler. Remarkable electrochemical and electrical properties were achieved, and the graphitic carbon obtained in this study is another important step towards achieving graphitic carbon from biomass. Hence this study presents the importance of type of lignin and carbonization process and how they impact on the final properties of carbon materials. This work gives new insights towards the effective utilization and value addition of the natural resources, reducing environmental problems, developing economy, and in-creasing the overall sustainability.”

Comment: Authors should justify the importance of the current work of how it is different from earlier reports. So, it’s better to add comparison table material and its performance to show the importance of the manuscript.

Response: Thank you, this study is the comparison of technical lignin as raw materials for making sustainable carbon. Study resulted in very interesting outcome that if energy storage is the end application, kraft lignin should be chosen as the initial raw material while if the need is to make graphitic or conductive carbon particles, soda lignin could be the right choice. Since different studies choose different measurement conditions and processing methods, direct comparison of the properties is difficult. Relevant comparisons are already included in the main text while explaining the electrochemical performance. 

Following change has been made in the manuscript: Page number 15, Table 7 has been added for comparing the high electrical conductivity obtained for SL1400 compared to lignin-based carbon particles produced in similar manner.

Materials

Carbonization temperature

(°C)

Electrical conductivity (S m─1)

Reference

SL1400

1400

335

This study

Carbonized lignin particles

900

0.9

Snowdon et.al, 2014

Carbon fillers

2000

142

Gindl-Altmutter et. al, 2015

L-900

900

33.3

Liu et.al, 2019

Comment: Please add the EIS analysis (Nyquist plot) for different samples.

Response: Thank you for the comment, we agree that analyzing EIS spectra and electrochemical performance of other carbons could be interesting, but we decided to analyze the electrochemical performance only of the KL1400 carbon particles which showed the highest specific surface area. The electrochemical impedance spectroscopy of KL1400 supercapacitors is shown in Figure 7d and discussed in the main text, page 14, line number 447 to 455.

Round 2

Reviewer 2 Report

This is a research article, however the title used for this research can not summarize what has been done. I suggest the authors revise the title so as to make it clear to the readers what has been done in this research.

Author Response

Comments to the Author 

Comment: This is a research article, however the title used for this research cannot summarize what has been done. I suggest the authors revise the title so as to make it clear to the readers what has been done in this research.

Response: Respected reviewer, we are thankful for your positive and constructive feedback. We have revised the title of the manuscript so that readers can get clearer idea about what  has been done in the study. New modified title is “Near 2D Carbon Structures from Sulfur-free Lignins for Functional Electrical and Electrochemical Devices”. In addition, we have polished the English language. We wish the revision to be satisfactory to the reviewer.